# Homology in Sex Determination in Two Distant Spiny Frogs, *Nanorana quadranus* and *Quasipaa yei*

**DOI:** 10.3390/ani14131849

**Published:** 2024-06-21

**Authors:** Yu Xiao, Guangjiong Liao, Wei Luo, Yun Xia, Xiaomao Zeng

**Affiliations:** 1Chengdu Institute of Biology, Chinese Academy of Sciences, Chengdu 610041, China; xiaoyu@cib.ac.cn; 2University of Chinese Academy of Sciences, Beijing 100049, China; 3Xiaozhaizigou National Nature Reserve, Beichuan, Mianyang 622750, China; lgjiong@163.com; 4Ecological Security and Protection Key Laboratory of Sichuan Province, Mianyang Normal University, Mianyang 621000, China; luoweihuyao@163.com

**Keywords:** sex chromosome, sex determination, homology

## Abstract

**Simple Summary:**

Many transitions of sex determination and sex chromosomes are found in amphibians. However, there is still controversy about how frequently such transitions have occurred. We carried out genotyping-by-sequencing and bioinformatics analyses on two closely related frogs (*Nanorana quadranus* and *Quasipaa yei*). The results show that both species involve male heterogamety concerning chromosome 1 and suggest that this chromosome would have been independently co-opted for sex determination in deeply divergent groups of anurans.

**Abstract:**

Sex determination is remarkably diverse, with frequent transitions between sex chromosomes, in amphibians. Under these transitions, some chromosomes are more likely to be recurrently co-opted as sex chromosomes, as they are often observed across deeply divergent taxa. However, little is known about the pattern of sex chromosome evolution among closely related groups. Here, we examined sex chromosome and sex determination in two spiny frogs, *Nanorana quadranus* and *Quasipaa yei*. We conducted an analysis of genotyping-by-sequencing (GBS) data from a total of 34 individuals to identify sex-specific makers, with the results verified by PCR. The results suggest that chromosome 1 is a homologous sex chromosome with an XY pattern in both species. This chromosome has been evolutionarily conserved across these closely related groups within a period of time. The *DMRT1* gene is proposed to be implicated in homology across two distantly related spiny frog species as a putative candidate sex-determining gene. Harboring the *DMRT1* gene, chromosome 1 would have been independently co-opted for sex determination in deeply divergent groups of anurans.

## 1. Introduction

In sharp contrast with mammals and birds, most amphibians are homomorphic with little-differentiated sex chromosomes in both sexes, and sex determination mechanisms are labile and diverse [1,2,3]. Indeed, most of the sex determination mechanisms reported in vertebrates, such as the XX/XY male heterogametic system in mammals and the ZZ/ZW female heterogametic system in birds, have been revealed in amphibians, and different sex-determination systems can even coexist in species of the same genus or in one species with different geographic populations [4,5,6,7,8,9,10,11]. Some recent research explains this diversity by sex chromosome transition in which sex-determining genes are translocated or replaced by another gene from a different chromosome [12,13,14,15]. The sex-determining locus is often found on nonhomologous chromosomes in some closely related species, and it can even exhibit variation within a single species [8,16]. By placing the sex-determining systems for each species onto a phylogenetic tree, high frequency transitions have been detected in some clades of amphibians [17,18,19].

However, although sex chromosomes transitions have occurred multiple times in amphibians, they have not been randomly selected. Some empirical evidence indicates that some chromosomes seem to be recurrently co-opted (i.e., recruited) for sex chromosomes [20,21]. On the basis of sex linkage and karyotypic data, 7 out of all 13 pairs of chromosomes have been predicted to be candidate sex chromosomes (Chrs) in true frogs (i.e., Chrs 1, 2, 3, 4, 5, 7, and 9) [22,23,24,25,26,27]. Jeffries et al. (2018) identified sex-determining system and sex chromosomes in 28 species of true frogs by using restriction-site associated DNA sequencing (RAD-seq) [18]. They found that at least 13 turnover events among 28 species. Of 13 events, only 5 pairs of chromosomes are used as sex chromosomes, with 3 of them having emerged multiple times as sex chromosomes, whereas other 2 chromosomes never appeared as such. In pipid frogs, at least seven independently evolved sex determination mechanisms have been described across both ZZ/ZW and XX/XY [28,29]. Coincidentally, there are also five pairs of chromosomes used as sex chromosomes in pipid frogs based on the genomic location of *Xenopus tropicalis* [29].

Why are some chromosomes more likely to be recurrently co-opted for sex chromosomes? One possible explanation is that they harbor a series of particular genes that play key roles in sex determination [30,31,32,33]. The *DMW* gene (*DMRT1* paralog) identified in *X. laevis* was the only master sex determining gene known within the entire amphibian group [34]. Recently, the newly uncovered structural X/Y variation in the 5′-regulatory region of the *BOD1L* gene in *Bufo viridis* indicated a Y-specific candidate sex determination locus in amphibians [35]. Several sex-linked genes that have been linked to gonadal differentiation in different frogs have been identified on certain chromosomes [27,30,31,32,33]. Among the genes implicated in feminization, we find *CYP19*, *SF1*, *FOXL2*, *SOX3*, etc., while genes associated with masculinization include *DMRT1*, *AMH*, *AR*, *CYP17*, and so on. In frogs, these potential candidates with gene ontology related to sex determination/ sex differentiation, seemingly limited to few chromosomes out of the complement of 13 haploid sets, are *DMRT1* and *AMH* on Chr 1; *FGF9*, *AMHR2* and *RSPO1* on Chr2; *CYP19* and *FOXL2* on Chr 3; *SOX3*, *SF*, and *AR* on Chr 7; and *CYP17* on Chr 9 [19,27].

However, highly frequent transitions are limited to a few frog groups, mostly emerging across deeply divergent taxa like families or genera except across ranid and pipid species [18,29,36]. We need more data, especially between closely related groups to understand the pattern of sex chromosomes evolution in amphibians. Spiny frogs, belonging to the tribe Paini in the family Dicroglossidae, live mostly in swift boulder-strewn streams in the mountains of South and Southeast Asia. Most male frogs are characterized by keratinized spines on the chest, belly, or other parts of their bodies. This tribe consists of two main radiations, corresponding to the genera *Nanorana* and *Quasipaa*, which diverged 27 million years ago [37]. In this study, we identified sex-specific markers via genotyping-by-sequencing (GBS) in two spiny frogs, *Nanorana quadranus* and *Quasipaa yei*, to investigate the sex chromosomes and sex-linkage genes. We used these spiny frog species to address two questions: (1) Is there homology between sex determination in these closely related genera? If yes, these chromosomes are evolutionarily conserved or recurrently co-opted? (2) Are there candidate sex-determining genes involved in homology? And here, are these particular genes evolutionarily conserved or not?

## 2. Materials and Methods

### 2.1. Sampling and Preparation

A combined sample of 15 individuals comprising 7 females and 8 males were collected for *N. quadranus*, whereas a total of 19 individuals consisting of 9 females and 10 males were collected for *Q. yei* in 2019 (Appendix A). The physiological sex of samples was determined by their secondary sexual characteristics and histology. The swelled vent above the anus in adult males was thought to be a unique structure and a kind of secondary sexual characteristic in *N. quadranus* and *Q. yei*. In addition, keratinized spines near the anus are often found in adult males of *Q. yei* [38]. Muscular tissues were taken from both females and males and stored in 95% ethanol for subsequent analysis. Of these, 31 samples were used for PCR verification.

The frog sampling procedure was permitted and approved by the local ethics committee for Animal Care and Use Committee of Chengdu Institute of Biology (CIB), Chinese Academy of Sciences (Permit Number: CIB-2017009).

### 2.2. Genomic Library Preparation

Genomic DNA was isolated from muscle tissues using a Qiagen^®^ DNeasy Blood and Tissue Kit (QIAGEN, Valencia, CA, USA) according to the manufacturer’s instructions for animal tissue extraction. Briefly, the muscle tissues (500 mg) were frozen in liquid nitrogen, milled to powder, and then digested with proteinase K at 56 °C for 1 h. Genomic DNA was purified from 20 mg of tissue powder using a DNeasy Blood & Tissue Kit (QIAGEN, Valencia, CA, USA), and eluted in 100 µL of RNase-free water. Genomic libraries were constructed following Elshire et al. [39]. Initially, genomic DNA was digested with *MseI* restriction enzymes (New England Biolabs, NEB, Ipswich, MA, USA) at 37 °C, and individual barcodes were ligated to the restriction sites. Subsequently, the DNA samples were purified using a Gel Extraction Kit (QIAGEN, Valencia, CA, USA), and DNA fragments of approximately 300–350 bp were excised from agarose gels. Furthermore, the prepared library was sequenced on the Illumina NovaSeq6000 platform at Novogene Bioinformatics Technology Co., Ltd., Beijing, China (www.novogene.cn; accessed on 15 December 2019) using a 150 bp paired-end protocol. In the end, the original sequences, also known as raw data, were obtained.

### 2.3. Filtering and SNP Calling

The software package Stacks-2.41 was chosen for GBS data analysis due to its versatility and capacity to handle substantial de novo GBS datasets [40]. Initially, we employed the *process_radtag* tool within Stacks-2.41 to clean the raw GBS data, eliminating low-quality reads such as those missing the restriction site. Following this, we executed the *de novo_map* function of the Stacks program. In this stage, we utilized three critical parameters: the minimum depth of coverage (m, −m = 2), the maximum number of allowed mismatches within stacks (alleles) for an individual (M, −M = 2), and the maximum number of mismatches permitted between individuals (*n*, −*n* = 1).

### 2.4. Screening Sex-Linked Markers

Sex-linked markers are typically found on the heterogametic sex chromosome: Y-chromosome markers in species where males are heterogametic and W-chromosome markers in species where females are heterogametic. For screening sex-linked markers, particularly in the context of the XY sex-determination system, we used two approaches in accordance with Brelsford et al. [41]. The first approach is based on sex differences in heterozygosity. A locus was deemed sex-linked if it exhibited homozygosity in half of the females and heterozygosity in at least half of the male samples. The second approach is based on sex-specific loci that only appear in one sex. Loci are considered Y-specific if they are completely absent in females and present in at least half of males. It is important to note that these methods are generally reversed when applying them to species with a ZW sex-determination system. Additionally, we employed a bespoke R script, using R-4.1.1 (R Core Team 2022) [42,43], to identify sex-linked loci through association analysis of the output generated by Stacks.

### 2.5. Confirmation of Sex-Specific Markers

To eliminate false positives from the identified potential sex-linked markers, a two-step process was implemented. Initially, we used a Linux grep command to remove any putative sex-limited GBS tags that were present in the original read files of the opposite sex. Next, we utilized BLAST 2.12.0+ to further filter out false positives [44], creating a local BLAST database from the *population.sample.fa* file produced by Stacks. We then aligned the sex-linked loci identified by our R script to this database using BLAST 2.12.0+ [44]. Putative sex-linked loci with the highest match scores were considered significant and retained if their E-value was equal to or less than 1 × 10^−10^. For instance, in the context of the XY system, if a sex-linked locus aligned homogeneously with the female sequence and heterogeneously with the male sequence, it was considered a true positive and retained (the criteria were reversed for the ZW system).

### 2.6. PCR Validation

PCR and gel electrophoresis assays were designed for validating the confirmed sex-linked markers that were truly sex-linked. We designed two types of primers to perform PCR verification. Taking XY systems as an example, first, several reads exhibited multiple SNP sites. During primer design, both upstream and downstream primers were targeted at these two SNP sites, with the male SNP being designated as the first base at the 3′ end of the primer sequence. The second method involved designing primers based on male-specific sequences. Male-specific sequences refer to those that exist only in the male genome (Y chromosome). Therefore, by directly designing primers based on male-specific sequences, we amplified bands that are specific to males. The PCR products were separated by electrophoresis (1.5%), and we expected that a band could be seen in males rather than in females (see details in Appendix A). 

### 2.7. Assigning Sex-Linked Markers onto Sex Chromosome

Based on the close phylogenetic relationships, the *Quasipaa spinosa* genome and the *Nanorana parkeri* genome were selected as the reference genomes [45,46]. We used the direct mapping method to determine the chromosomal location of sex-linked markers. The sex-linked markers were directly mapped to the reference genome using BLAST [44] to determine relative positions of the sex-linked markers on the reference genome. The top matches with e-values of no more than 1 × 10^−10^ were retained.

### 2.8. Predicting the Sex-Linked Markers Involved in Genes for Sex Determination

Amphibians have highly diverse sex-determining modes, but the molecular mechanisms of sex determination are still unclear. According to previous research, candidate sex-determining genes in amphibians were compared, including *AMH*, *AMHR2*, *AR*, *CYP17*, *CYP19A1*, *RSPO1*, *BOD1L*, *DMRT1*, *FOXL2*, *SOX3*, and *SF1*. To verify whether the obtained sex-linked markers serve as candidate sex-determining genes, an indirect approach was utilized. 

The initial step involved aligning the coding sequences of the candidate sex-determining genes with the reference genome to locate their positions. Subsequently, we extended and trimmed the alignment regions and aligned them with the sex-linked loci to determine whether any sex-linked loci were located within those regions. Here, the *Q. spinosa* and *N. parkeri* genomes were selected as reference genomes based on their close phylogenetic relationships [45,46].

## 3. Results

### 3.1. GBS Data Analyses and SNP Calling

A combined total of 34,134,552,576 raw Illumina sequencing reads were acquired from a genotyping-by-sequencing (GBS) library prepared for 10 male and 9 female individuals of the species *Q. yei*. For *N. quadranus*, a total of 22,175,915,040 raw Illumina bases were obtained from a GBS library constructed for eight male and seven female *N. quadranus* individuals. Following the removal of low-quality sequences using the *process_radtags* tool, a total of 56,309,046,030 reads were retained for both species (Appendix A). Subsequently, the Stacks *de novo_map* pipeline was applied for SNP discovery, resulting in the identification of a catalogue comprising 1,260,290 SNPs across the populations of these two species.

### 3.2. Sex-Linked Marker Screening

The method based on heterozygosity differences revealed the presence of 33 sex-linked SNPs (located on 26 GBS tags) exhibiting an XY pattern in *N. quadranus* and 907 sex-linked SNPs (located on 630 GBS-tags) displaying an XY pattern in *Q. yei*. Notably, no sex-linked SNP displaying the ZW pattern was detected in either of these species. Leveraging the sex-limited occurrence approach, 267 male-limited GBS tags were identified in *N. quadranus*, while 1148 male-limited and 10 female-limited GBS tags were identified in *Q. yei*. In aggregate, a total of 2365 putative sex-linked markers were identified in these two species using the aforementioned approaches (Appendix A).

### 3.3. Validation of the Sex-Linked Markers

The screening process for sex-linked markers within our GBS datasets revealed markers exhibiting the anticipated XY or ZW patterns, suggesting the possibility of false positives. Following the elimination of false positives using BLAST, we identified 28 sex-linked loci (located on 24 GBS tags) and 66 sex-specific sequences with an XY pattern, resulting in a total of 94 confirmed sex-linked markers for *N. quadranus.* For *Q. yei*, a total of 598 sex-linked markers and 489 sex-specific markers were identified through this process. Ultimately, it was verified that *N. quadranus* and *Q. yei* exhibit an XY sex-determination system (Appendix A).

As with the SNP-based analysis, we found no male specific GBS tag shared between the species.

### 3.4. PCR Validation

For each species, we randomly selected some markers, which were validated to design two types of primers for the PCR test. Three markers were successfully detected by agarose gel electrophoresis after amplification, and only males exhibited bands. Ultimately, one sex-linked marker was validated for *Q. yei*, and two sex-linked ones were validated for *N. quadranus* (Figure 1 and Appendix A, Appendix A).

### 3.5. Identifying the Sex Chromosome

In this study, we directly mapped all sex-linked markers to the genome of *Q. spinosa* and *N. parkeri*. In *N. quadranus*, a total of 59 sex-linked markers were mapped to the genome of *Q. spinosa*, with 40 out of the 59 (68%) sex-linked markers being successfully mapped to chromosome 1; the remaining markers are distributed across the remaining 12 pairs of chromosomes. For *Q. yei*, a total of 981 sex-linked markers were successfully mapped to the genome of *Q. spinosa*, and 597 out of 981 (60.9%) of the sex-linked markers could be mapped to chromosome 1. The remaining 39.1% of the markers are scattered across the remaining 12 pairs of chromosomes. 

Using the *N. parkeri* genome as a reference genome, out of the 94 confirmed sex-linked markers of the *N. quadranus*, 64 sex-linked markers were successfully aligned to the reference genome, while the remaining markers could not be successfully aligned. Of the 64 sex-linked markers that were aligned successfully, 54 sex-linked markers concentrate on chromosome 1. For *Q. yei*, out of 1087 sex-linked markers, 697 sex-linked markers were successfully aligned to the *N. parkeri* reference genome. Among the 697 sex-linked markers successfully aligned, 50% of the markers are concentrated on chromosome 1, while the remaining markers are distributed across other chromosomes, ranging from 1% to 8%.

The reasons for the unsuccessful alignment of sex-linked markers may be that these markers are noncoding genes, exhibit significant differences, or lack homologous sequences in the reference genome. Regardless of using the genome of *Q. spinosa* or *N. parkeri* as the reference genome, the sex-linked markers of *N. quadranus* and *Q. yei* are both concentrated on chromosome 1. These findings indicated that in *N. quadranus* and *Q. yei*, chromosomes 1 might function as the sex chromosome pair, containing a collection of sex-linked loci (Figure 2 and Figure 3, Appendix A). 

### 3.6. Identifying Potential Sex-Determining Genes

Based on the indirect method, only the *DMRT1* gene of *G. ruguosa* is located on chromosome 1 of the reference genome. The remaining candidate sex-determining genes either failed to align to the reference genome or were aligned to multiple chromosomes. 

By using the *Q. spinosa* genome as a reference genome, it was found that both *N. quadranus* (CLoci_2513909, 2462691, 2385711) and *Q. yei* (CLoci_2989553, 2989074, 2920011) possess three sex-linked markers positioned within the alignment region of *DMRT1* on the reference genome. Conversely, upon utilizing the *N. parkeri* genome, it was revealed that *N. quadranus* contains three sex-linked markers (CLoci_2513909, 2462421, 2450800), while *Q. yei* exhibited same markers (CLoci_2989553, 2989074, 2920011) situated in the alignment region of *DMRT1* on the reference genome (Figure 4; see details in Appendix A.

Taking into account the accumulated evidence, we propose that *DMRT1* is the most probable candidate sex-determining gene in *N. quadranus* and *Q. yei*.

## 4. Discussion

### 4.1. Sex Determination Is Limited to Homologous Chromosomes

Our results show that the majority of sex-linked markers with male heterogamety from both spiny frogs were mapped to the genome of *Q. spinosa* as well as *N. parkeri*, both corresponding to chromosome 1 (Figure 2 and Figure 3, Appendix A). Chromosome 1 was then co-opted as the sex chromosome with an XY system in both species. And thus, the sex determination was limited to homologous chromosomes in two distant spiny frogs, where they belong to closely related genera, *Quasipaa* and *Nanorana*, diverged 27 million years ago [37].

Chromosome 1 turns out to be sex-linked one in these two closely related genera, *Quasipaa* and *Nanorana*. In fact, our additional examinations revealed a consistent XY system in other investigated species, with chromosome pair 1 being shared in these species of *Quasipaa* and *Nanorana* (Xia, Xiao, Zeng, unpublished). Thus, our findings so far support that the species from *Quasipaa* and *Nanorana* retain an ancestral sex chromosome pair, which has been evolutionarily conserved for 27 million years. 

Many investigated cases in frogs show that the species of closely related groups shared the same sex-linked chromosome pair. In the European tree frog clade, including four species of the *Hyla arborea* group, the Tyrrhenian *Hyla sarda*, and the Middle Eastern *Hyla savignyi*, all species share the same sex chromosome pair within approximately 11 M years [49,50,51]. The largest chromosome pair with an XY system was retained in all diploid species of the *Bufo viridis* subgroup, including *B. siculus*, *B. shaartusiensis*, *B. balearicus*, *B. turanensis*, *B. variabilis*, *B. viridis*, and probably *B. boulengeri* [52]. Similarly, chromosome 5 was sex-limited with male heterogamety across at least eight species of the western clade of torrent frog *Amolops* within approximately 31 M years [53]. In such cases, transitions of sex chromosomes have happened infrequently across closely related species or groups, and they would then have been evolutionarily conserved within a period of time.

Moreover, species sharing a homologous sex chromosome pair have been discovered even across deeply divergent groups. Chromosome 1 was proved to be extensively conserved synteny across four anuran families (Pipidae, Ranidae, Hylidae, and Bufonidae) within approximately 210 million years [36]. This chromosome appears to be sex-linked across the species of some anuran families, i.e., Ranidae, and Hylidae [36,51], which now extends to spiny frogs of the family Dicroglossidae (this study). The more plausible explanation may be that this chromosome has independently evolved sex linkage multiple times in these groups over more than 160 million years (http://timetree.org; accessed on 4 February 2024). Through sex chromosome transitions have been found in many deeply divergent clades in amphibians, the high frequency of transitions in closely related genera or species is not common, except in some ranid and pipid species [17,19]. For example, within the genus *Rana*, sex chromosome transitions have happened multiple times, and chromosome 1 (corresponding to *Xenopus tropicalis* chromosome 1) has been recurrently co-opted as the sex chromosome in at least eight species [18]. In this situation, the possibility that species from these different families retain an ancestral sex chromosome pair might be excluded. When sex chromosome transitions are frequently biased toward certain chromosomes, this bias would become a self-reinforcing evolutionary process. When a chromosome has been sex-linked in the past, it might have accumulated genes likely to be concerned with sexually antagonistic effects, which could then make it more likely to recapture the role of a sex chromosome in a turnover event [54,55]. Thus, chromosome 1 being homologous sex-linked has been recurrently co-opted as the sex chromosome across these families. 

Notably, despite very few cases, sex chromosomes seem to be evolutionarily conserved among closely related groups within a period of time. Across deeply divergent clades when estimated on a large evolutionary scale of taxa, considering the high frequency of sex chromosome transitions, some chromosomes are more likely to have been recurrently co-opted for sex chromosomes. 

### 4.2. Sex Determination May Pertain to the DMRT1 Gene

In the present study, the transcription factor *DMRT1* of the frog *G. rugosa* was the gene that was identified that matched with sex-linkage markers in both spiny species investigated (Figure 3 and Figure 4, Appendix A). Furthermore, there is no other candidate gene that was found involving in these sex-linked markers in both species. The *DMW* gene (*DMRT1* paralog) is the only one detected gene among potential candidates known in frogs [34]. Thus, the candidate sex-determining genes would be involved in the homology between two distant spiny frogs.

The *DMRT1* gene is possibly harbored by chromosome 1, which was found to be sex-linked in both spiny frogs of the family Dicroglossidae (Figure 3 and Figure 4, Appendix A). This corresponds to the records in *Rana* frogs of the Ranidae and *Hyla* tree frogs of the Hylidae, where sex-linked *DMRT1* has been suggested to be located on chromosome 1 [36,49,51,56]. Ranid frogs together with those of the Rhacophoridae and Mantellidae, have been grouped into the sister clade of Dicroglossidae, which diverged 75 million years ago [48]. Accordingly, considerable synteny is shown among frog genomes, despite long periods of independent evolution [45,57,58]. The *DMRT1* gene would then have the same chromosomal location across three families over more than 160 million years (Figure 3) (http://timetree.org; accessed on 4 February 2024) [48].

What might have predisposed chromosome 1 to repeatedly evolving sex linkage in these deeply divergent taxa? The presence of the *DMRT1* gene might be a good explanation. This gene appears to concern the male differentiation pathway throughout the whole animal kingdom, from flies to mammals. The *DMRT1* gene or its paralogs determine sex in birds, medaka fish, and African clawed frogs, making it an appealing candidate gene for sex determination in species where it is sex-linked. Harboring the *DMRT1* gene, chromosome 1 would have been independently co-opted for sex determination in deeply divergent groups over a long period of time.

## 5. Conclusions

XY sex-determination systems were detected in both *N. quadranus* and *Q. yei* species by GBS and verified by PCR. The results of mapping sex-linked markers onto the genome of *Q. spinosa* suggest that chromosome 1 serves as a sex chromosome in both species. This finding indicates that this chromosome seems to be evolutionarily conserved across these closely related groups over 27 million years. The transcription factor *DMRT1* was found to be a putative candidate sex-determining gene implicated in homology in two distantly related spiny species. Furthermore, harboring the *DMRT1* gene, chromosome 1 has been independently co-opted for sex determination in deeply divergent groups over a long period of time. The patterns of sex chromosome evolution in spiny frogs imply that transitions are not highly frequent in many groups of frogs.

## Figures and Tables

**Figure 1 animals-14-01849-f001:**
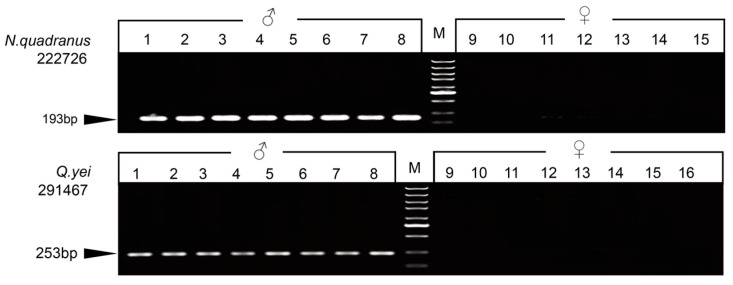
Gel electrophoresis showing the PCR amplification of markers 22276 (*N. quadranus*) and 291467 (*Q. yei*). The locus ID is indicated to the left. ‘M’ indicates a DNA marker. Black arrows indicate the PCR product size.

**Figure 2 animals-14-01849-f002:**
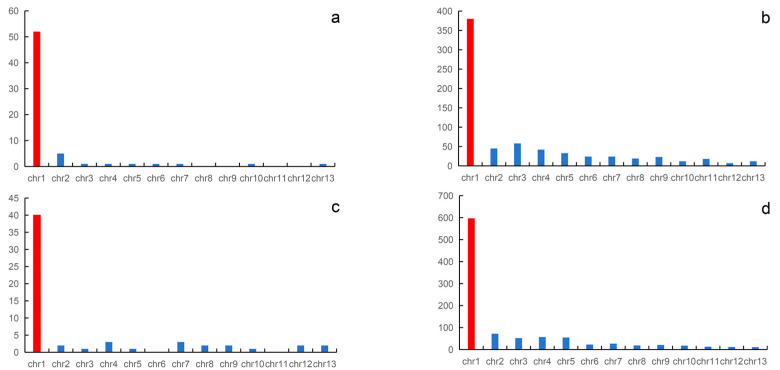
Mapping of the sex-linked markers of *N. quadranus* and *Q. yei* to the reference genome. The sex-linked markers of *N. quadranus* and *Q. yei* are mapped to the *N. parkeri* genome (**a**,**b**). The sex-linked markers of *N. quadranus* and *Q. yei* are mapped to the *Q. spinosa* genome (**c**,**d**), respectively. The red columns indicate sex chromosomes.

**Figure 3 animals-14-01849-f003:**
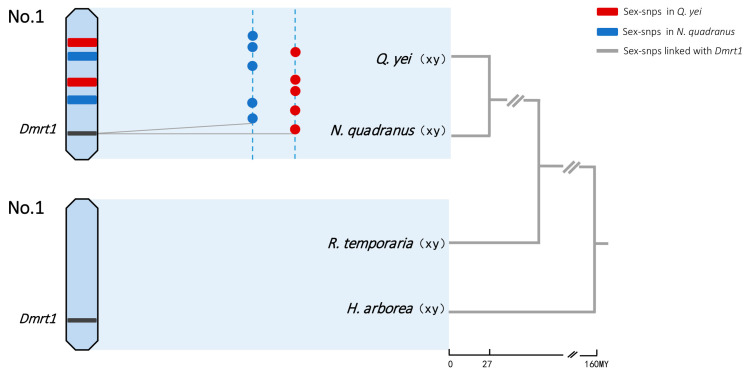
Sex chromosomes and sex-linked genes in four frogs. Sex-determination system and sex-chromosome identities were from GBS data in *N. quadranus* and *Q. yei* (the majority of sex-linked markers mapped to chromosome 1 of *Q. spinosa* and *N. parkeri*: see detail in Appendix A, Appendix A) and from the literature in *R. temporaria* and *H. arborea* [36,47]. Chromosome 1 harbors the candidate gene *Dmrt1* for sex determination in four frogs. The dotted lines represent the sex chromosome for each species, and colored dots show sex-linked SNPs in *N. quadranus* (blue) and *Q. yei* (red). Sex-linked SNPs are randomly located, and there was no shared XY site found. The grey solid lines show that several sex-linked SNPs were mapped to the *Dmrt1* gene. Tree topology and approximate divergence times (My) were adapted using a combination of data from Che et al. (2010), Yuan et al. (2019) [37,48] and http://timetree.org (accessed on 4 February 2024).

**Figure 4 animals-14-01849-f004:**
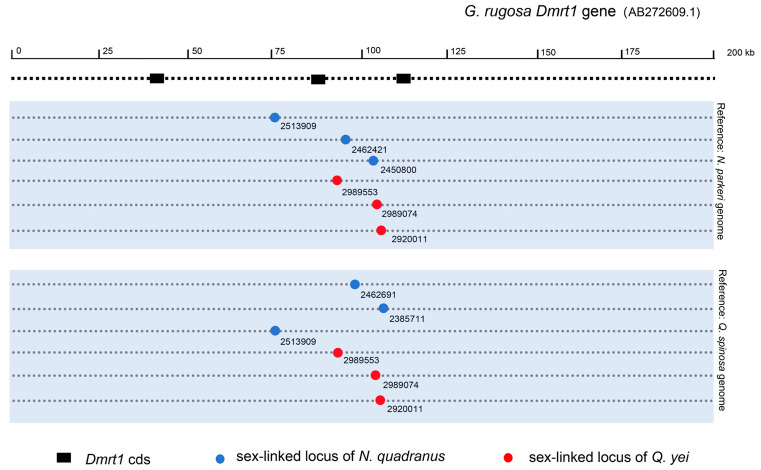
The mapping of sex-linked locus of *N. quadranus* and *Q. yei* on *DMRT1* gene of *G. rugosa*. Circles indicate sex-linked loci of *N. quadranus* and *Q. yei*, and dashed lines indicate the relative alignment region of the *DMRT1* gene of *G. ruguosa* in the reference genome. The blue background sections display the reference genomes utilized for screening sex-linked loci.

## Data Availability

The data supporting the results of this study can be found in the manuscript. The raw sequence data obtained in this study were deposited in the China National Center for Bioinformation and are publicly accessible at CRA015437.

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
