# Peer review of "Homology in Sex Determination in Two Distant Spiny Frogs, *Nanorana quadranus* and *Quasipaa yei"

_animals, 2024, doi:10.3390/ani14131849_

Round 1

Reviewer 1 Report

Comments and Suggestions for Authors

The proposed manuscript (ms) ‘Homology in sex-determination in two distant spiny frogs’ significantly contributes to the study of sex chromosome evolution in frogs. The ms focuses on two spiny frog species from the family Dicroglossidae, identifies sex chromosomes, and proposes Dmrt1 as a candidate gene involved to sex determination in both species. The structure of the ms is well designed. The ms is technically sound, and presented in an understandable fashion. Scientific names are well used. Appropriately chosen terms and abbreviations are clearly explained and used for better understanding. Used methodology is on a solid level. I have comments and suggestions related mainly to the introduction and discussion sections. A major shortcoming I see is that the authors overlooked one of the most used groups of anurans, pipid frogs, which involves two model organisms - Xenopus tropicalis and X. laevis. There are many studies that explored sex chromosomes, sex loci, sex-specific SNPs. Importantly, only one master sex determining gene, dmw, is known within the entire amphibian group and this gene was identified in X. laevis (Yoshimoto et al 2008). At least seven independently evolved sex determination mechanisms were described in Pipidae across both ZZ/ZW and XX/XY (Cauret et al 2020) but the diversity of these mechanisms in this focal group is even higher (unpublished data). Thus I do not agree with the statement: “However, previous cases are very limited to a few frog groups, where mostly emerging across deeply divergent taxa like families or genera except across ranids species.”. Also, the sentence “In frogs, these potential candidates with a gene ontology related to sex determination/sex differentiation, seemingly limited on only few chromosomes out of the complement of 13 haploid sets, are suggested as DMRT1 and AMH on Chr 1; CYP19 and FOXL2 on Chr 3; SOX3, SF1 and AR on Chr 7; and CYP17 on Chr 9 [25]” do not contains all candidates genes and all sex chromosomes identified. Again in pipids, chromosomes 2, 3, 4, 6, 7, 8 with unknown master sex determining gene (it can be for example the ar gene on chromosome 8L in X. borealis) were identified as sex chromosomes. Another candidate master gene is bod1l localized on chromosome 1 in bufo viridis (Kuhl et al 2024, full text available on biorxiv.org).

Minor comments:

1) line 32: sex determination is complicated? I do not understand, please rephrase the sentence.

2) line 33: “ZZ/ZW female system” add “heterogametic”.

3) line 36: “Even different sex determination systems can co-exist in species of the same genus or in one species with different geographic populations.” Two Xenopus citations fit exactly with this sentence. In Furman et al (2021) three sex chromosomes were found in wild populations of X. tropicalis. In Song et al (2021) gene expression of sex-linked transcript corresponds to three different sex chromosomes in Xenopus species.

4) line 38: “of” replaced by “from”.

5) define co-option after the first use.

6) Describe a group of spiny frogs in the introduction.

7) line 113-115: “A locus was deemed sex-linked if it exhibited homozygosity in half of the females and heterozygosity in at least half of the male samples.” Based on what hypothesis do the authors conclude this condition? Can they add a citation where this hypothesis has been used?

8) line 119: can the R script be available for readers? For example on github.

9) line 147: reference genomes in plural instead of singular?

10) line 156: list of candidate genes is not complete

11) 330-331: “Thus, the candidate sex-determining genes would be the same gene in two distant spiny frogs.” I am not sure with this claim, there is no evidence.

12) line 336: DMRT1 in Bufo toads of Bufonidae is not a candidate according to the recent study Kuhl et al (2024).

14) The conclusions section is repetition of results.

References:

Cauret CMS, Gansauge MT, Tupper AS, Furman BLS, Knytl M, Song XY, Greenbaum E, Meyer M, Evans BJ. Developmental Systems Drift and the Drivers of Sex Chromosome Evolution. Mol Biol Evol. 2020 Mar 1;37(3):799-810. doi: 10.1093/molbev/msz268. Erratum in: Mol Biol Evol. 2020 Jun 1;37(6):1844. PMID: 31710681.

Furman BLS, Cauret CMS, Knytl M, Song XY, Premachandra T, Ofori-Boateng C, Jordan DC, Horb ME, Evans BJ. A frog with three sex chromosomes that co-mingle together in nature: Xenopus tropicalis has a degenerate W and a Y that evolved from a Z chromosome. PLoS Genet. 2020 Nov 9;16(11):e1009121. doi: 10.1371/journal.pgen.1009121. PMID: 33166278; PMCID: PMC7652241.

Kuhl H, Tan WH, Klopp C, Kleiner W, Koyun B, Ciorpac M, Feron R, Knytl M, Kloas W, Schartl M, Winkler C, Stöck M. A candidate sex determination locus in amphibians, evolved by structural variation between X- and Y-chromosomes. Nat Commun (accepted). 2024. preprint https://www.biorxiv.org/content/10.1101/2023.10.20.563234v2.

Song XY, Furman BLS, Premachandra T, Knytl M, Cauret CMS, Wasonga DV, Measey J, Dworkin I, Evans BJ. Sex chromosome degeneration, turnover, and sex-biased expression of sex-linked transcripts in African clawed frogs (Xenopus). Philos Trans R Soc Lond B Biol Sci. 2021 Aug 30;376(1832):20200095. doi: 10.1098/rstb.2020.0095. Epub 2021 Jul 12. PMID: 34247503; PMCID: PMC8273505.

Yoshimoto S, Okada E, Umemoto H, Tamura K, Uno Y, Nishida-Umehara C, Matsuda Y, Takamatsu N, Shiba T, Ito M. A W-linked DM-domain gene, DM-W, participates in primary ovary development in Xenopus laevis. Proc Natl Acad Sci U S A. 2008 Feb 19;105(7):2469-74. doi: 10.1073/pnas.0712244105. Epub 2008 Feb 11. PMID: 18268317; PMCID: PMC2268160.

Comments on the Quality of English Language

English needs a revision: wording, phrases, typos, missing spaces. Some expressions such as “here” are not clear where is “here”.

Reviewer 2 Report

Comments and Suggestions for Authors

Comments about the manuscript:

“Homology in sex-determination in two distant spiny frogs”

Sex determination in amphibians frequently involves chromosome shifts. Some chromosomes that become sexual exist recurrently in divergent taxa. However, the evolution of sex chromosomes in closely related groups is still poorly understood. In the study presented here, the authors examined the sex chromosomes of two spiny frogs, Nanorana quadranus and Quasipae yei in relation to sex determination. To do this, they analyzed genotyping by sequencing (GBS) data verified by PCR from 34 individuals. The results show that chromosome 1 is the homologous sex chromosome with an XY pattern in both species and that the DMRT1 gene is involved in the homology between the two species.

This genetic analysis work provides results contributing to the understanding of the genetic aspect of sex determination in amphibians, a question which is not yet completely resolved. The methodology seems well done to me, the conclusions convincing. However, I propose some ideas for improving the manuscript.

Page 1, title: add the scientific names of the two species studied in the title.

Page 2, materials and methods, line 7. “A total of 34 adults…”: Give additional information on the equipment used. For each species, specify how many males and how many females were studied, specify the size of the animal studied, and what secondary sexual characteristics were taken into account.

Page 2, line 89. “according to the manufacturer's instructions for animal tissue extraction.” Briefly describe the method: the phrase "according to the manufacturer's instructions" does not seem sufficient to me for a scientific article.

Page 6, lines 147 and 164: reference 38 is before reference 37. To be renumbered continuously.

Page 11, lines 488-489. I did not find reference 47 in the text.

Page 11, lines 497-498. This is not a reference: delete this sentence (which is written below).

Round 2

Reviewer 1 Report

Comments and Suggestions for Authors

The proposed manuscript ‘Homology in sex-determination in two distant spiny frogs, Nanorana quadranus and Quasipaa yei’ has been revised and improved. The authors followed my recommendations and I believe that the article will be of more interest to the readers than the original version of the manuscript.  The manuscript meets the requirements for a scientific publication in the “Animals” journal. I am looking forward to seeing the published version.

Comments on the Quality of English Language

English is acceptable.
